

# Between extreme simplification and ideal optimization: antennal sensilla morphology of miniaturized *Megaphragma* wasps (Hymenoptera: Trichogrammatidae)

Anna V. Diakova, Anastasia A. Makarova and Alexey A. Polilov

Department of Entomology, Biological Faculty, Moscow State University, Moscow, Russia

## ABSTRACT

One of the major trends in the evolution of parasitoid wasps is miniaturization, which has produced the smallest known insects. *Megaphragma* spp. (Hymenoptera: Trichogrammatidae) are smaller than some unicellular organisms, with an adult body length of the smallest only 170 µm. Their parasitoid lifestyle depends on retention of a high level of sensory reception comparable to that in parasitoid wasps that may have antennae hundreds of times larger. Antennal sensilla of males and females of *Megaphragma amalphitanum* and *M. caribea* and females of the parthenogenetic *M. mymaripenne* are described, including sensillum size, external morphology, and distribution. Eight different morphological types of sensilla were discovered, two of them appearing exclusively on female antennae. Two of the types, sensilla styloconica and aporous placoid sensilla, have not been described previously. Regression analyses were performed to detect and evaluate possible miniaturization trends by comparing available data for species of larger parasitoid wasps. The number of antennal sensilla was found to decrease with the body size; *M. amalphitanum* males have only 39 sensilla per antenna. The number of antennal sensilla types and sizes of the sensilla, however, show little to no correlation with the body size. Our findings on the effects of miniaturization on the antennal sensilla of *Megaphragma* provide material for discussion on the limits to the reduction of insect antenna.

## INTRODUCTION

Antennae are present in all insects and perform various functions, the most obvious being perception of sensory information (*Schneider, 1964*). In parasitoid wasps they function to find the host habitat and location, and evaluate host's condition and suitability for oviposition (*Van Baaren et al., 2007*), and are involved in courtship and mating behavior (*Bin et al., 1989*). Parasitoid wasps antennae often have a high diversity of antennal sensilla, e.g., 14 different morphological types are found on female antenna of *Trichogramma australicum* Girault, 1912 (Hymenoptera: Trichogrammatidae) (*Amornsak, Cribb & Gordh, 1998*). Sensilla numbers can also be exceptionally high, e.g., *Microplitis croceipes* (Cresson,

Corresponding authors
Anna V. Diakova,
anndiakova@yandex.ru
Alexey A. Polilov,
polilov@mail.bio.msu.ru,
polilov@gmail.com

1872) (Hymenoptera: Braconidae) individuals have about 9,000 sensilla per antenna (*Das et al., 2011*)).

One of the principal directions in parasitoid wasp evolution is miniaturization, producing such peculiar organisms as the strongly reduced male of *Dicopomorpha echmepterygis Mockford, 1997* (Hymenoptera: Mymaridae), the smallest known adult insect, with a body size of 139–240 μm (*Mockford, 1997*). Such a decrease in body size alters their morphology, physiology, ecology and behavior (*Eberhard & Wcislo, 2011*; *Polilov, 2016*). Their sensory organs also undergo a number of size-related adaptations. Thus, miniaturization of an eye in the smallest parasitoid wasps results in the reduction of ommatidium and rhabdom lengths and other changes in ommatidium structure (*Fischer, Müller & Meyer-Rochow, 2011*; *Makarova, Polilov & Fischer, 2015*). Reduction in number of antennomeres and number of antennal sensilla in the smallest insect species also occurs (*Polilov, 2015*; *Polilov, 2017*), and a correlation of antennae size with the body size was shown in Chalcidoidea (*Symonds & Elgar, 2013*). But there have been no studies specifically on the miniaturization of antennal sensilla in parasitoid wasps.

Some of the smallest parasitoid wasps belong to *Megaphragma* (Hymenoptera: Trichogrammatidae); their body sizes are around 200 μm. Miniaturization affects them at the level of organs, cells, and even cellular structures; e.g., adults have unique anucleate neurons (*Polilov, 2012*). While adult *Megaphragma* retain the complexity of internal structure, they also have reductions or losses, such as absence of the heart and considerable reductions in the set of muscles and tracheal system (*Polilov, 2017*), features common to the smallest insects. As egg-parasitoids of the greenhouse thrips, *Heliothrips haemorrhoidalis* (Bouché, 1833) (Thysanoptera: Thripidae) (*Bernardo & Viggiani, 2002*), female *Megaphragma* successfully perform the same kinds of behavior important for the survival of any parasitoid species, including detection of the host eggs, which are embedded under the cuticle of leaves (*Del Bene, Gargani & Landi, 1998*), and recognition of their host at a species-specific level. Their antennae evidently remain highly functional and sensitive, despite being extremely small.

Antennal sensory structures of larger parasitoid wasps have been thoroughly studied in earlier publications. The antennal sensilla of more than 30 species of 10 different families have been investigated based on SEM photographs, with TEM studies used in some studies to discover their inner ultrastructure (*Van Baaren et al., 2007*). Within-species variations of sensillum sizes, numbers of sensilla, and their morphology is often described, and were thoroughly studied in *Trichogramma evanescens* Westwood, 1833 (Hymenoptera: Trichogrammatidae) (*Van der Woude & Smid, 2016*). Most of the studies treat Braconidae (*Xi et al., 2010*; *Zhou et al., 2011*) and Chalcidoidea, with special emphasis on Trichogrammatidae, Scelionidae and Platygastridae (*Cave & Gaylor, 1987*; *Olson & Andow, 1993*; *Amornsak, Cribb & Gordh, 1998*; *Isidoro, Romani & Bin, 2001*; *Zhang et al., 2012*). Some studies covered the biology and behavior associated with the antennal sensilla in parasitoid wasps, making it possible to predict the functions of some types of sensilla (*Norton & Vinson, 1974*; *Schmidt & Smith, 1986*). Despite inconsistency in terms and definitions of types of sensilla, and the shift of the focus of studies mostly to female specimens, the data obtained on the subject provides us with an opportunity to make

assumptions on evolutionary trends in the antennal sensilla morphology of parasitoid wasps.

The aim of this work was to study the effects of miniaturization on the antennal sensilla of *Megaphragma*, and estimate the limits to the reduction of sensilla in a functional antenna.

## MATERIALS AND METHODS

### Material

Adult *Megaphragma amalphitanum* Viggiani, 1997, *Megaphragma mymaripenne* Timberlake, 1924 and *Megaphragma caribea* Delvare, 1993 were reared from eggs of *H. haemorrhoidalis*. For the study of gross morphology, specimens were fixed in alcoholic Bouin; for the study of sensilla ultrastructure, specimens were fixed in 2% glutaraldehyde (GA, EMS) solution in sodium 0.1 M cacodylate buffer pH = 7.2 with subsequent postfixation in 1% osmium oxide (EMS) solution in the same buffer.

### Scanning electron microscopy

The fixed material was gradually dehydrated through a series of ethyl alcohols (GA fixed material—30%, 50%, 70%, 95% ethyl alcohol, each change for 30 min, 100%—two changes for 30 min; alcoholic Bouin fixed material—same protocol starting from the 70% ethyl alcohol step) and then acetone (100%, two changes for 15 min), critical point dried (Hitachi HCP-2) and sputtered with gold (Giko IB-3). The specimens were studied and photographed using Jeol JSM-6380 with a 5 megapixel digital camera and FEI Inspect F50 with a 4 megapixel digital camera.

### Morphometry

All measurements were performed on the SEM images, using measurement tool in the Fiji package of ImageJ. Normality test, descriptive statistic, ANOVA, SMA and OLS were performed using R software.

## RESULTS

### *Megaphragma amalphitanum* antennae

Antennae of *M. amalphatanum* are geniculate at the scape-pedicel joint (Fig. 1). Male and female antennae are almost identical in shape and composition, containing the following antennomeres: scape including radicula, pedicel, and flagellum consisting of 1 ringlike anellus and three flagellomeres, the 1st flagellomere forming the funicle and the 2nd and 3rd flagellomeres forming the club (sometimes this structure is also called "clava") (Fig. 2). Thus, antenna of this species is composed of five segments with two pronounced sub-segments, the radicula and anellus. The radicula forms the base of the scape and is inserted in the head via torulus, the basal socket joint upon which the radicula is articulated and which allows antennal movement in all directions. Antennal segments are elongated and cylindrical; the last flagellomere is slightly tapered. Female antennae measure $144 \pm 9 \, \mu m$ (mean $\pm$ sd) in length; male antennae are $136 \pm 10 \, \mu m$ long (Table S1).

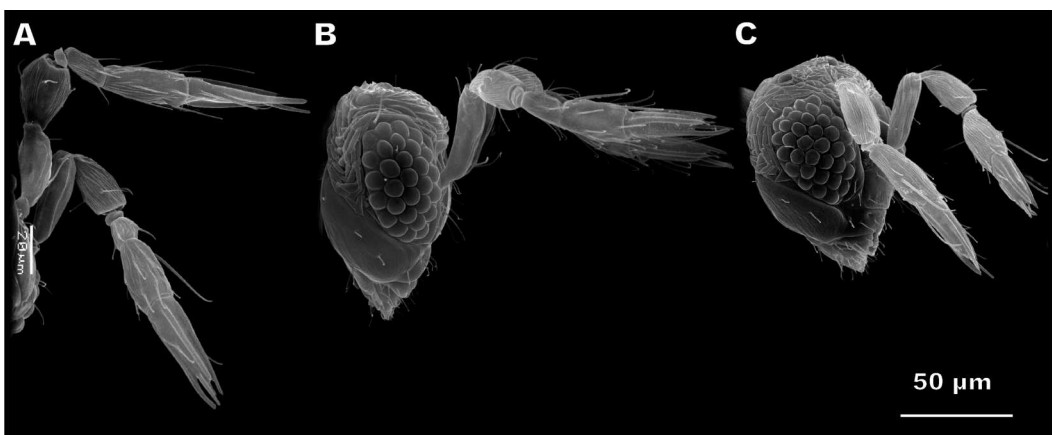

**Figure 1** Female antennae of *Megaphragma amalphitanum* (A), *M. mymaripenne* (B) and *M. caribea* (C) (SEM, lateral view).

### Morphological types of antennal sensilla and their distribution

The total number of antennal sensilla was invariable between specimens and was found to be 43 sensilla per female antenna and 39 sensilla per male antenna in all of the 25 studied specimens. We distinguished eight distinct types of sensilla on male and female antennae of *M. amalphatanum*: aporous sensilla chaetica (ChS-AP), aporous sensilla trichodea, type 1 (TS1-AP), aporous sensilla trichodea, type 2 (TS2-AP), uniporous sensilla trichodea (TS-UP), sensilla styloconica (SS), multiporous placoid sensilla (MPS), sensilla basiconica (BS), and aporous placoid sensilla (PS-AP), the two latter are unique to the female antennae. The morphology, numbers and distribution of antennal sensilla was invariable within species.

### Sensilla chaetica, aporous (ChS-AP)

These sensilla are tapered to a blunt aporous tip and inserted in a cuticular socket. Their surface is longitudinally fluted and bears no pores. They are typically protruding (Fig. 3B). Three ChS-AP appear on the scape, five on pedicel and one on the 1st flagellomere in the female antenna. Overall disposition remains the same for the male specimens, except for the 1st flagellomere, where one more sensillum was observed (Fig. 2). ChS-AP are 11.5 ± 2.62 µm in length and 0.9 ± 0.16 µm in diameter in females and 9.89 ± 2.88 µm in length, 0.71 ± 0.11 µm in diameter in males (Table 1). According to the differences in sizes on different segments of the antenna, ChS-AP may be further divided into subtypes. In females, chaetica sensilla on the 1st flagellomere are significantly longer than on the pedicel and scape. By contrast, males had the longest and thickest ChS-AP on the scape, whereas difference between ChS-AP on the 1st flagellomere and pedicel was insignificant. When comparing sizes between sexes, we estimated that ChS-AP on the 1st flagellomere are significantly longer and thicker in females (Table S2).

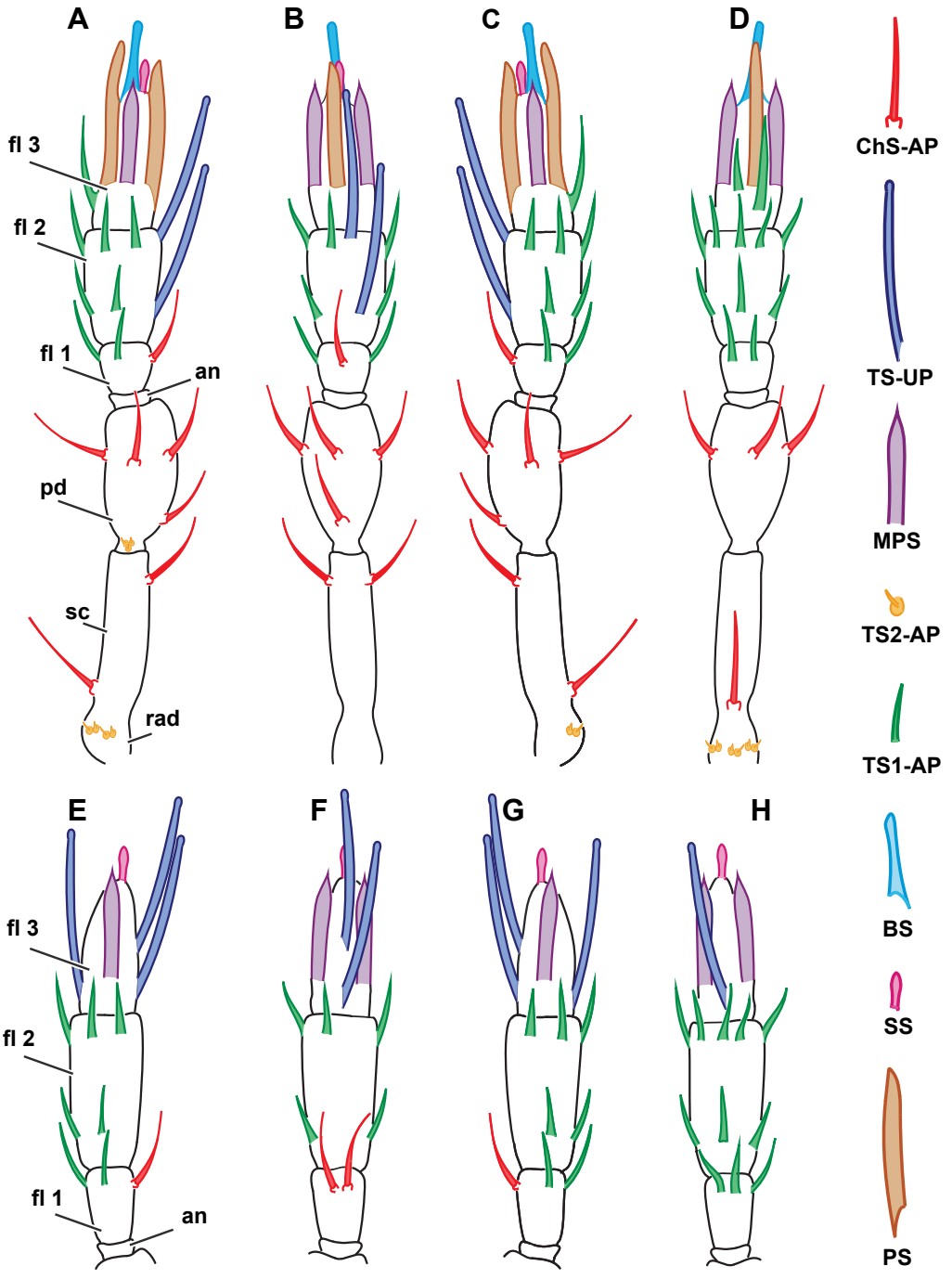

**Figure 2 Distribution of antennal sensilla in *Megaphrgama amalphitanum*.** Antennae are unnaturally straightened in this figure; for their natural shape, see Fig. 1. (A) female, medial view; (B) female, dorsal view; (C) female, lateral view; (D) female, ventral view; (E) male, medial view; (F) male, dorsal view; (G) male, lateral view; (H) male, ventral view. Sc, scape; rad, radicula; pd, pedicel; an, anellus; fl 1, 1st flagellomere; fl 2, 2nd flagellomere; fl 3, 3rd flagellomere.

 

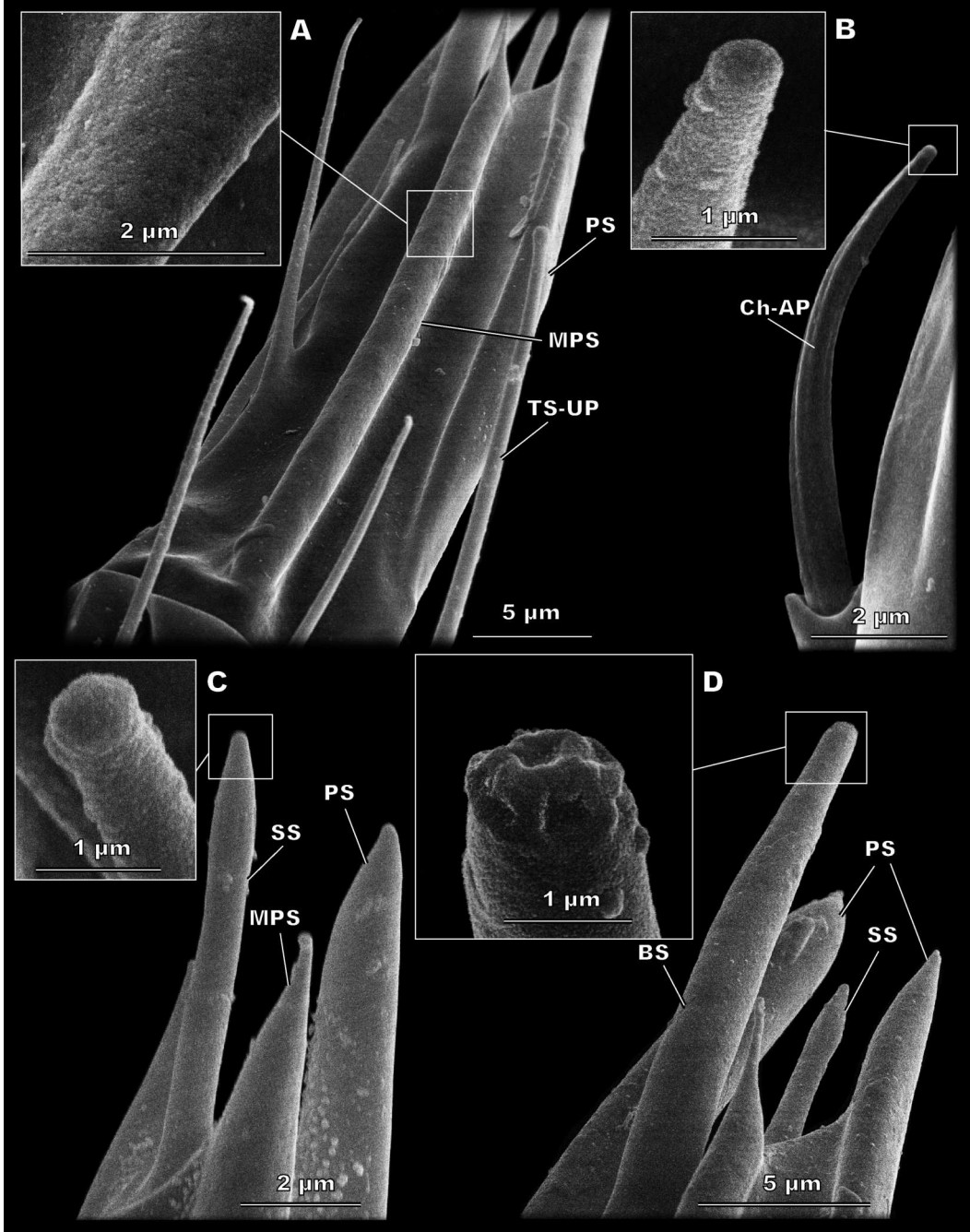

**Figure 3   Morphological types of antennal sensilla in *Megaphragma amalphitanum* (SEM).** (A) Female club, demonstrating PS, MPS with an elongated tip and TS-UP with a widened uniporous tip. Pores on the MPS wall can be seen on a close-up. (B) ChS-AP with a socket and a fluted wall and a close-up of its aporous tip. (C) Sole SS situated on the apex of antenna with aporous tip. (D) Sole BS, unique to females, multiple tip pores are observed on a close-up.

**Table 1** *Megaphragma amalphitanum, M. mymaripenne* and *M. caribea* numbers per antenna and sizes of antennal sensilla (mean ± sd), measurements are given in μm. For additional details, see Table S3.

| Sensilla type | Parameter | *Megaphragma amalphitanum* | | *M. mymaripenne* | *M. caribea* | |
| | | Female | Male | Female | Female | Male |
| --- | --- | --- | --- | --- | --- | --- |
| ChS-AP | Number | 9 | 10 | 9 | 8 | 8 |
| | Length | 11.5 ± 2.62 | 9.89 ± 2.88 | 14.2 ± 2.03 | 8.04 ± 1.49 | 6.28 ± 0.99 |
| | Diameter | 0.9 ± 0.16 | 0.71 ± 0.11 | 1.3 ± 2.68 | 0.75 ± 0.12 | 0.67 ± 0.13 |
| TS1-AP | Number | 17 | 14 | 22 | 18 | 25 |
| | Length | 12.7 ± 5.42 | 7.63 ± 1.59 | 13.68 ± 4.52 | 8.41 ± 3.54 | 5.16 ± 1.29 |
| | Diameter | 0.82 ± 0.14 | 0.65 ± 0.09 | 0.85 ± 0.14 | 0.72 ± 0.13 | 0.62 ± 0.12 |
| TS-UP | Number | 2 | 3 | 2 | 2 | 3 |
| | Length | 37.1 ± 5.59 | 33.6 ± 4.46 | 29.93 ± 4.78 | 33.03 ± 5.97 | 20.42 ± 2.87 |
| | Diameter | 1.36 ± 0.23 | 1.34 ± 0.16 | 1.36 ± 0.19 | 1.19 ± 0.12 | 1.28 ± 0.2 |
| SS | Number | | | | | |
| | Length | 7 ± 0.72 | 5.05 ± 0.48 | 5.78 ± 1.48 | 4.78 ± 0.83 | 4.58 ± 1.2 |
| | Diameter | 0.93 ± 0.11 | 0.76 ± 0.09 | 0.86 ± 0.2 | 0.99 ± 0.11 | 1 ± 0.15 |
| TS2-AP | Number | 9 | 9 | 9 | 9 | 9 |
| | Length | 0.88 ± 0.22 | 1.03 ± 0.36 | 0.93 ± 0.24 | 0.89 ± 0.21 | 0.97 ± 0.23 |
| | Diameter | 0.55 ± 0.14 | 0.53 ± 0.12 | 0.51 ± 0.07 | 0.57 ± 0.03 | 0.61 ± 0.27 |
| BS | Number | 1 | – | 1 | 1 | – |
| | Length | 17.1 ± 2.04 | – | 14.8 ± 2.19 | 11.77 ± 2 | – |
| | Diameter | 1.96 ± 0.19 | – | 1.85 ± 0.29 | 1.53 ± 0.21 | – |
| PS | Number | 2 | – | 2 | 2 | – |
| | Length | 39.9 ± 4.15 | – | 29.77 ± 3.84 | 43,324 | – |
| | Diameter | 2.48 ± 0.6 | – | 2.22 ± 0.51 | 1.87 ± 0.19 | – |
| | Tip length | 11.7 ± 4.32 | – | 9.17 ± 4.08 | 6.83 ± 1.86 | – |
| MPS | Number | 2 | 2 | 2 | 2 | 2 |
| | Length | 37.3 ± 3.58 | 21.9 ± 5.61 | 31.94 ± 3.57 | 25.74 ± 2.29 | 21.55 ± 2.5 |
| | Diameter | 2.62 ± 0.27 | 1.82 ± 0.38 | 2.45 ± 0.5 | 2.3 ± 0.15 | 1.73 ± 0.35 |
| | Tip length | 3.69 ± 1.36 | 3.76 ± 4.51 | 6.66 ± 4.77 | 1.8 ± 1.16 | 21.55 ± 2.5 |

### *Sensilla trichodea, aporous type 1(TS1-AP)*

TS1-AP are slender, with an acute tip and a smooth surface lacking grooves and pores. They arise directly from the antenna without any kind of socket or rim, and usually are directed along its surface (Fig. 4C). It is the most abundant type of sensilla in both sexes. In females, four are found on the 1st, eleven on the 2nd, and two on the 3rd flagellomeres. Males have three TS1-AP on the 1st flagellomere and eleven on the 2nd flagellomere; they lack TS1-AP on the 3rd flagellomere (Fig. 2). TS1-AP measure 12.7 ± 5.42 μm in length and 0.82 ± 0.14 μm in diameter in females and 7.63 ± 1.59 μm in length, 0.65 ± 0.09 μm in diameter in males (Table 1). In both sexes, the sensilla are significantly longer on each subsequent flagellomere (sensilla on the 1st Fl. <2nd Fl. <3d Fl. for females and 1st Fl. <2nd Fl. for males). Females demonstrated significantly thicker and longer TS1-AP on the 1st and 2nd flagellomeres than males (Table S2).

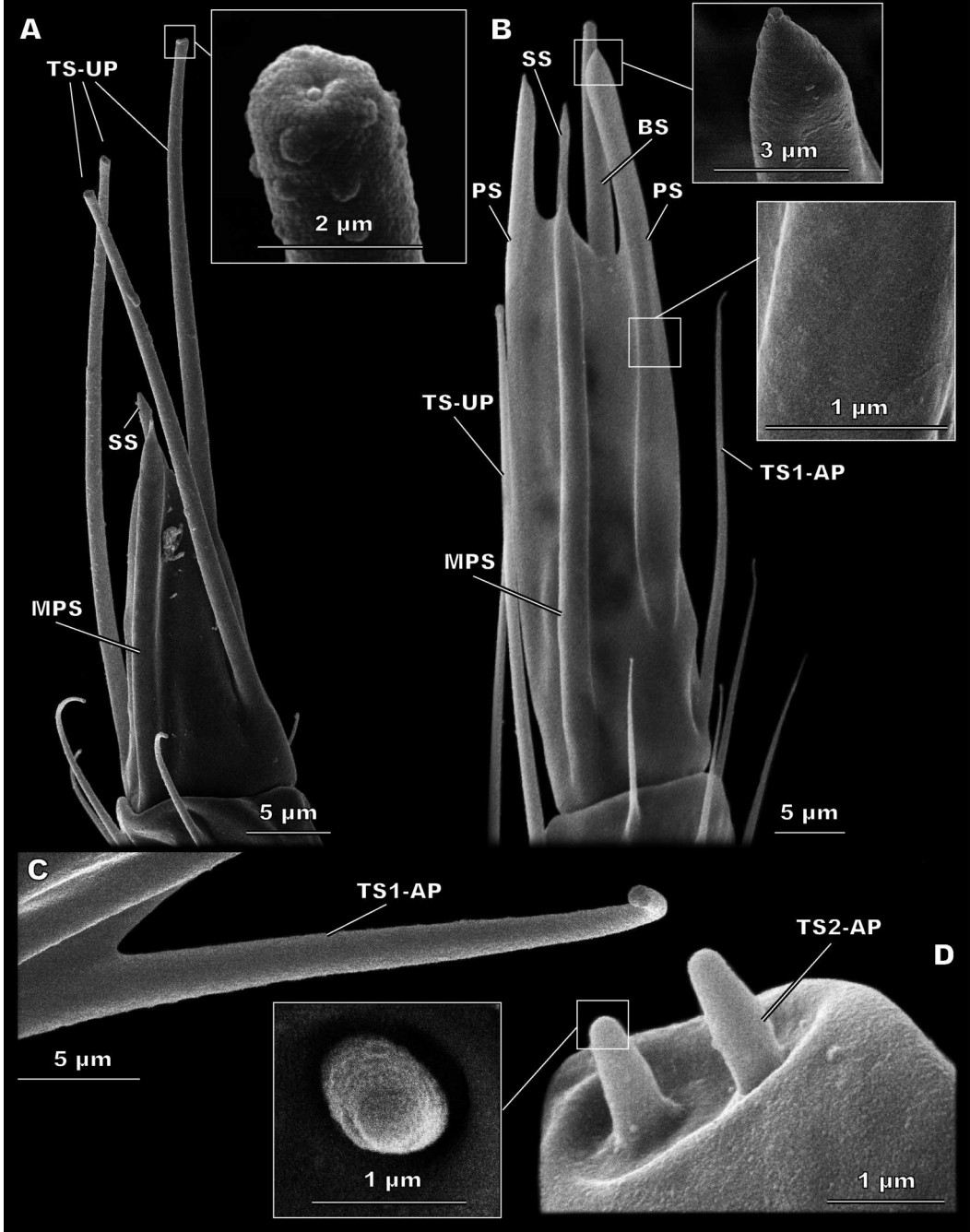

**Figure 4** **Morphological types of antennal sensilla in *Megaphragma amalphitanum* (SEM) (continuation).** (A) Male club with two MPS (one is situated on the other side of antennomere), sole SS and three TS-UP. The widened, uniporous tip of TS-UP can be seen on a close-up. (B) Female club, lateral view, demonstrating MPS, TS-UP, SS, PS and BS, two latter unique to females. Close-ups of PS demonstrate absence of pores on its tip and wall. (C) TS1-AP with aporous wall and tip. (D) TS2-AP with a close-up of its aporous tip.

### Sensilla trichodea, aporous type 2 (TS2-AP)

These short, aporous sensilla are tapered, with a blunt apex, and are surrounded by a circular rim (Fig. 4D). Their number and position are identical in both sexes: six TS2-AP occur in two bundles on the radicula, near the head-scape joint, and three on the pedicel, near the scape-pedicel joint (Fig. 2). TS2-AP are the smallest sensilla on the antenna, only $0.88 \pm 0.22 \,\mu m$ in length and $0.55 \pm 0.14 \,\mu m$ in diameter in females and $1.03 \pm 0.36 \,\mu m$ in length, $0.53 \pm 0.12 \,\mu m$ in diameter in males (Table 1). TS2-AP are significantly longer on the scape than on the pedicel, and in females also thicker, in both species. According to the intersex comparisons, females have shorter TS2-AP on the scape than males (Table S2).

### Sensilla trichodea, uniporous (TS-UP)

TS-UP are long and slender, longitudinally tapered, with a slightly widened tip and an apical pore. The wall surface is smooth and lacks pores; sometimes a shallow depression can be observed at the base. TS-UP are mainly aligned with the axis of the antenna (Fig. 4A). Two TS-UP appear at the 2nd flagellomere in females and three at the 3rd flagellomere in males, protruding far below the tip of the male antenna (Fig. 2). They measure $37.1 \pm 5.59 \,\mu m$ in length and $1.36 \pm 0.23 \,\mu m$ in diameter in females and $33.6 \pm 4.46 \,\mu m$ in length, $1.34 \pm 0.16 \,\mu m$ in diameter in males (Table 1). No significant size differences were found between sexes (Table S2).

### Sensilla styloconica (SS)

These bulb-shaped structures have a small-elongated tip and no pores (Fig. 3C). They are relatively small and there is only one of them per antenna, situated apically on the 3rd flagellomere (Fig. 2). They are significantly longer ($7 \pm 0.72 \,\mu m$) and wider ($0.93 \pm 0.11 \,\mu m$) in females than in males ($5.05 \pm 0.48 \,\mu m$ length and $0.76 \pm 0.09 \,\mu m$ width, respectively) (Table 1, Table S3).

### Sensilla basiconica (BS)

BS are found only in females. It is a robust structure situated apically at the tip of antenna and aligned with its axis with multiple pores on a blunt tip (Fig. 3D). BS protrude beyond and below all other sensilla on the 3rd flagellomere of the female antenna. BS are $17.1 \pm 2.04 \,\mu m$ long and $1.96 \pm 0.19 \,\mu m$ wide (Table 1).

### Multiporous placoid sensilla (MPS)

The walls of these long, thick sensilla are merged with the surface of the antenna and covered with numerous pores, observable at a high magnification ($9,000\times$) (Fig. 3A). The tips of the MPS are detached from the antennal surface and have small tapered aporous protrusions. MPS are situated identically in males and females. There are two of MPS per antenna; they appear at the lateral and median surfaces of the 3rd flagellomere, opposing each other and parallel to the longitudinal axis of the antenna (Fig. 2). They are significantly longer ($37.3 \pm 3.58 \,\mu m$) and wider ($2.62 \pm 0.27 \,\mu m$) in females than in males ($21.9 \pm 5.61 \,\mu m$ length and $1.82 \pm 0.38 \,\mu m$ width, respectively) (Table 1, Table S2). The detached tips measure $3.69 \pm 1.36 \,\mu m$ in females and $3.76 \pm 4.51 \,\mu m$ in males (Table 1).

### Placoid sensilla (PS)

These large, conspicuous sensilla are unique to females and occur on the 3rd flagellomere, opposing each other at its dorsal and ventral surfaces (Fig. 4B). No pores were observed even at a magnification of 130,000×. PS are fused with the antennal surface and form large detached protrusions at the tip of antenna (Fig. 2). They measure 39.9 ± 4.15 μm in length and 2.48 ± 0.6 μm in width; the protrusions are 11.7 ± 4.32 μm long (Table 1).

### *Megaphragma caribea* antennae

This species demonstrates fused flagellomeres in both sexes, without any visible edges between the 1st and 2nd flagellomeres and retaining only remnants of them between the 2nd and 3rd flagellomeres (Fig. 5). Additionally, the pedicel of males appears to be identical to that of females. Otherwise, however, the antennae of have a similar overall shape and composition to those of *M. amalphitanum* (Fig. 1). The male and female antennae are 120 ± 9 μm and 120 ± 7 μm long, respectively (Table S1).

### Morphological types of antennal sensilla and their distribution

There are 44 sensilla on the female antenna and 49 on the male antenna (12 studied specimens). The morphological types of sensilla are identical to those in *M. amalphitanum* (Fig. 6). No difference was observed between the two species in overall sensilla structure, presence of socket, or sensilla surface properties, such as grooves and pores. The numbers and distribution of antennal sensilla in are similar to those of *M. amalphitanum*, except as follows. The fused 1st and 2nd flagellomeres lack ChS-AP in both males and females. Females have 18 TS1-AP whereas males have 25. Both sexes have an additional SS on the distal edge of the 2nd flagellomere (Figs. 5 and 6). It appears that this sensillum is significantly shorter than SS on the 3rd flagellomere in females (Table S2). The sizes of the antennal sensilla and the results of ANOVA are provided in the tables (Table 1, Tables S2–S4).

### *Megaphragma mymaripenne* antennae

Since *M. mymaripenne* is a parthenogenetic species, the data was obtained for the females of this species only. Antennae of *M. mymaripenne* are identical in shape and composition to those of *M. amalphitanum* females (Fig. 1) and are 140 ± 9 μm long (Table S1).

### Morphological types of antennal sensilla and their distribution

49 sensilla were found on the antenna in *M. mymaripenne* (15 specimens studied). The antennal sensilla morphology and types of sensilla are identical to those of *M. amalphitanum* females. 22 aporous TS1-AP were found. An additional SS was seen on the distal edge of the 2nd flagellomere, significantly longer than SS on the 3rd flagellomere (Figs. 5 and 6). Except for the above-mentioned differences, the overall distribution and numbers of antennal sensilla were identical to *M. amalphitanum* females. Sizes of *M. mymaripenne* antennal sensilla and the results of ANOVA are provided in the tables (Table 1, Tables S2–S4).

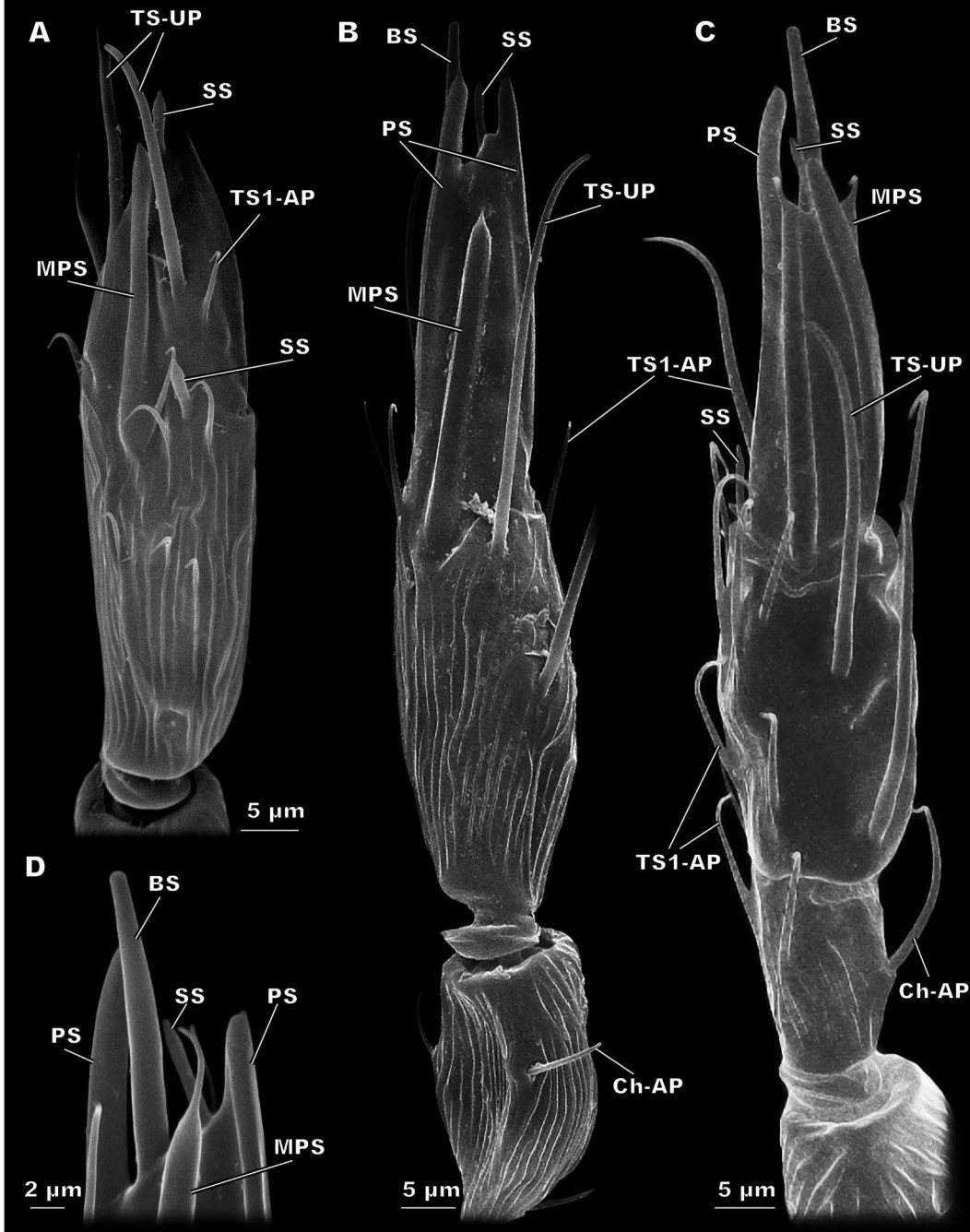

**Figure 5** **Detailed view of the antenna of *Megaphragma caribea* and *M. mymaripenne* (SEM).** (A) Male
*M. caribea* fused flagellum demonstrating numerous TS1-AP, MPS, TS-UP and two SS. (B) Female *M.
caribea* antenna with three fused flagellomeres. (C) Female *M. mymaripenne* antenna. (D) Tip of *M. my-
maripenne* antenna.

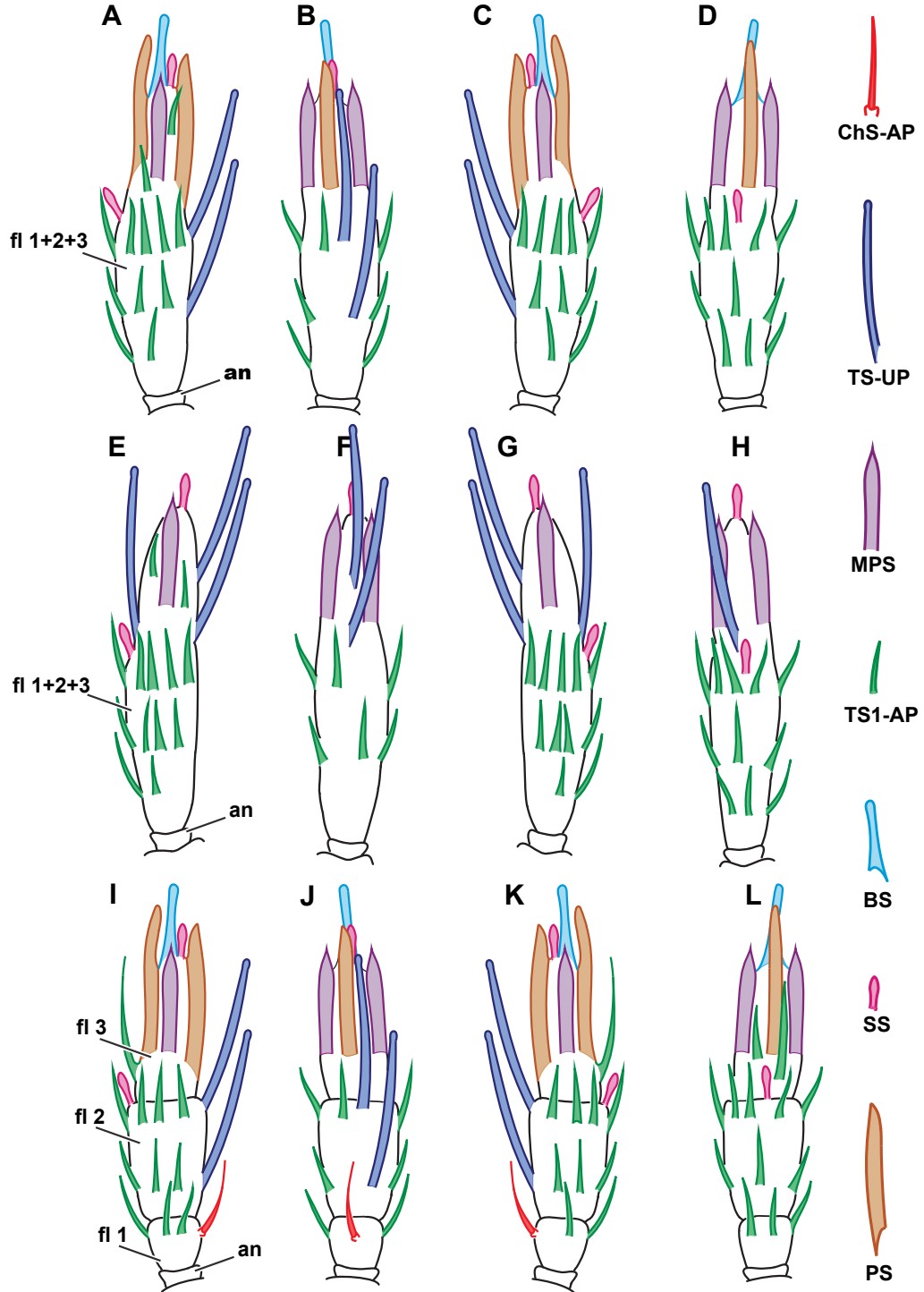

**Figure 6  Distribution of antennal sensilla in *Megaphragma caribea* and *M. mymaripenne*.** (A) Female *M. caribea*, medial view; (B) female *M. caribea*, dorsal view; (C) female *M. caribea*, lateral view; (D) female *M. caribea*, ventral view; (E) male *M. caribea*, medial view; (F) male *M. caribea*, dorsal view; (G) male *M. caribea*, lateral view; (H) male *M. caribea*, ventral view; (I) female *M. mymaripenne*, medial view; (G) female *M. mymaripenne*, dorsal view; (K) female *M. mymaripenne*, lateral view; (L) female *M. mymaripenne*, ventral view. An, anelli; fl 1, 1st flagellomere; fl 2, 2nd flagellomere; fl 3, 3rd flagellomere.

## DISCUSSION

### Antennal size and gross morphology

The overall composition of antennae in *M. amalphitanum*, *M. caribea* and *M. mymaripenne* is typical for most groups of parasitoid hymenopterans. All three species have three flagellomeres (fused in *M. caribea* males and females), as in the females of several *Trichogramma* species (*Olson & Andow, 1993*; *Amornsak, Cribb & Gordh, 1998*; *Consoli, Kitajima & Parra, 1999*; *Zhang et al., 2012*; *Van der Woude & Smid, 2016*). Most Trichogrammatidae and Mymaridae, and most larger parasitoid wasps in other families of parasitoid wasps have significantly more flagellar segments; e.g., *Pteromalus puparum* (Linnaeus, 1758) (Hymenoptera: Pteromalidae) has 12 flagellomeres, and *M. croceipes* and *Cotesia marginiventris* (Cresson, 1865) (Hymenoptera: Braconidae) have 16 flagellomeres (*Das et al., 2011*). The observed reduction in the number of flagellomeres is a common feature among miniaturized insects, known also in some of the smallest species of Ptiliidae and Corylophidae (Coleoptera) (*Polilov, 2015*).

The antennae of *M. caribea* are considerably different from those of *M. mymaripenne* and *M. amalphitanum*: in *M. caribea* flagellomeres are fused in both sexes (Figs. 1C, 6A–6H). In female egg parasitoids, the last flagellomeres are sometimes fused into a club (*Consoli, Parra & Zucchi, 2010*), e.g., in *Amitus spiniferus* (Brèthes, 1914) (Hymenoptera: Platygasroidea) (*Isidoro, Romani & Bin, 2001*). By contrast, *Ooencyrtus phongi* Trjapitzin, Myartseva & Kostjukov, 1977 (Hymenoptera: Encyrtidae), *Metaphycus parasaissetiae* Zhang and Huang, 2007 (Hymenoptera: Encyrtidae), *Trichogramma dendrolimi* Matsumura, 1926 and *T. australicum* Girault, 1912 (Hymenoptera: Trichogrammatidae) have a single club segment in males, which is separated into flagellomeres in females (*Amornsak, Cribb & Gordh, 1998*; *Xi et al., 2011*; *Zhang et al., 2012*; *Zhou et al., 2013b*). In the case of *M. caribea*, both sexes have a fused club, which is uncommon.

The antennal lengths of the three species are diminutive, from 120 μm in *M. caribea* to 144 in *M. amalphitanum* females (Table S1), and minute Chalcidoidea species have antennae that are 1.5–4 times longer (*Amornsak, Cribb & Gordh, 1998*; *Van Baaren et al., 1999*; *Zhou et al., 2013a*; *Zhou et al., 2013b*). Interestingly, in the shortest winged insect, the parasitoid wasp *Kikiki huna* Huber, 2000 (Hymenoptera: Mymaridae) (*Huber & Noyes, 2013*), antennal length is more than 151 μm, which makes the antennae of *Megaphragma* possibly the smallest functional parasitoid antennae ever described.

### Number of antennal sensilla

The number of antennal sensilla significantly decreases with the body size in parasitoid wasps (Fig. 7A, Table 2). *T. australicum* have 3–6 times as many sensilla as the studied species (*Amornsak, Cribb & Gordh, 1998*), Pteromalidae species have 23 times as many as the studied species (*Onagbola & Fadamiro, 2008*; *Dweck, 2009*), and some Braconidae species have hundreds of times as many (*Gao, Luo & Hammond, 2007*; *Xi et al., 2010*; *Das et al., 2011*). The number of antennal sensilla recorded in the studied species is the smallest known among parasitoid wasps, except for the highly reduced male of *Dicopomorpha echmepterygis* (Hymenoptera: Mymaridae) (*Mockford, 1997*). Even the minute *K. huna*, the smallest winged insect (body length = 158–190 μm) appears to have about twice as

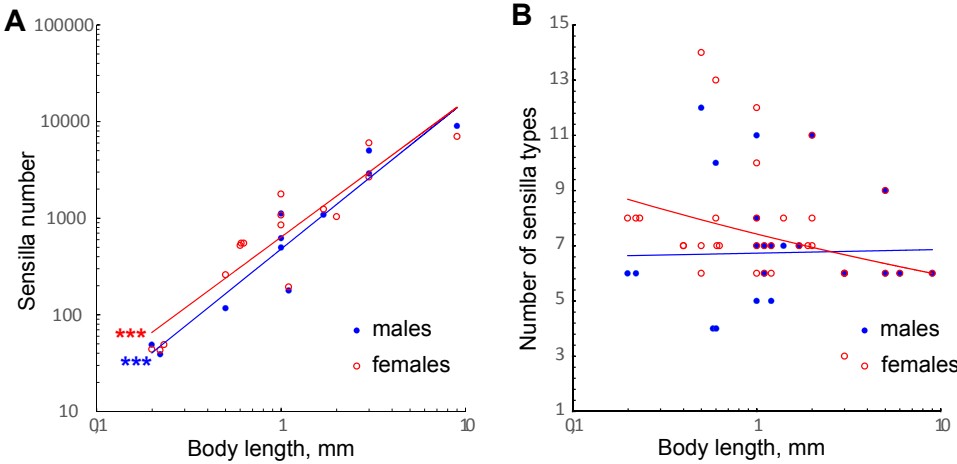

**Figure 7  Changes in number of antennal sensilla and number of types of antennal sensilla in parasitoid wasps.** (A) The effect of miniaturization on the number of antennal sensilla. Data on Chalcidoidea and Braconidae species are included. For references, see Table S4. (B) Number of antennal sensilla types as a function of body length. Data on Chalcidoidea and Braconidae species are included. For references, see Table S4. Trendlines representing correlated variables ($p < 0.01$) with the slopes of SMA and OLS regressions significantly different from 0 ($p < 0.01$) are marked with three asterisks (***); for slope values and $R^2$ see Table 2.

**Table 2  Slope values and $R^2$ of regression analyses performed on the data on the number of antennal sensilla, number of antennal sensilla types, and antennal sensilla sizes in parasitoid wasps.**

| Dependent variable/independent variable | SMA | | OLS | |
| --- | --- | --- | --- | --- |
| | Slope | $R^2$ | Slope | $R^2$ |
| Number of antennal sensilla/body length, males | 1.60[*] | 0.91 | 1.53[*] | 0.91 |
| Number of antennal sensilla/body length, females | 1.53[*] | 0.84 | 1.4[*] | 0.84 |
| Number of antennal sensilla types/body length, males | 0.30 | 0.0007 | 0.008 | 0.0007 |
| Number of antennal sensilla types/body length, females | 0.24 | 0.10 | 0.08 | 0.11 |
| ChS-AP length/body length, females | 0.45 | 0.007 | 0.037 | 0.007 |
| ChS-AP diameter/body length | 0.42 | 0.35 | 0.25[**] | 0.19 |
| MPS length/body length | 0.64 | 0.36 | 0.39[**] | 0.36 |
| MPS diameter/body length | 0.76 | 0.26 | 0.39 | 0.11 |
| TS2-AP length/body length | 0.94[*] | 0.56 | 0.71[*] | 0.56 |
| TS2-AP diameter/body length | 0.58[**] | 0.45 | 0.39[**] | 0.45 |

**Notes.**
[*]Variables are correlated, slope is significantly different from 0 with $p < 0.01$.
[**]Variables are correlated, slope is significantly different from 0 with $0.01 \leq p < 0.05$.

many antennal sensilla as the species treated in our study, judging by SEM photographs (*Huber & Noyes, 2013*). Thus, it is very likely that *M. amalphitanum*, *M. mymaripenne* and *M. caribea* are nearing the lowest number of antennal sensilla required for a highly sensitive parasitoid antenna.

The variation in number of sensilla also seems to be reduced, the smaller the species becomes. Thus, in the three species studied the numbers of antennal sensilla are completely

invariable within species (sexual dimorphism being taken into account). By contrast, in larger parasitoid wasps the numbers of sensilla vary significantly within species; e.g., in *O. phongi* (*Xi et al., 2011*) the number of trichoid sensilla of type 1 is 1,549 ± 67.9 (mean ± se, $n = 6$). Even in the tiny *Trichogramma australicum, T., galloi* Zucchi, 1988, *T. pretiosum* Riley, 1879, *T. nubilae* Ertle and Davis, 1974, and *T. evanescens* (Hymenoptera: Trichogrammatidae) the numbers of antennal sensilla significantly vary within species, and for several types of sensilla the correlation with the body size was shown (*Olson & Andow, 1993*; *Amornsak, Cribb & Gordh, 1998*; *Consoli, Kitajima & Parra, 1999*; *Van der Woude & Smid, 2016*).

## Morphological types of antennal sensilla

The terminology used to describe insect antennal sensilla is inconsistent and confusing. For instance, there are at least four terms describing one of the antennal sensilla types in *Trichogramma* (*Ruschioni et al., 2012*), one based on the external appearance of the sensillum ("Falcate Sensilla", *Amornsak, Cribb & Gordh, 1998*, "Multiporous Pitted Sensilla Trichoid C", *Olson & Andow, 1993*), another on its supposed functions ("Multiporous Gustatory Sensilla", *Isidoro et al., 1996*), and in one case these sensilla are designated with a letter ("Sensilla type i", *Voegelé et al., 1975*). In this study, we define and name sensilla types based on the morphological characteristics of sensilla as they appear when observed under a scanning electron microscope (SEM). The comparison of the terms adopted in this study with the terminology of earlier studies on parasitoid wasps is given in Table S5.

It is common for chalcidoid wasp males to have 1–3 types of sensilla fewer than females (*Barlin, Vinson & Piper, 1981*; *Amornsak, Cribb & Gordh, 1998*; *Van Baaren et al., 1999*; *Xi et al., 2011*; *Zhang et al., 2012*; *Zhou et al., 2013b*; *Namikawa & Amornsak, 2016*), although in several cases the same number of types was reported for both sexes (*Onagbola & Fadamiro, 2008*; *Jun et al., 2013*). It is likely that the additional types of sensilla found in females are required for the biological tasks exclusive for that sex, e.g., searching for host eggs and evaluating their suitability for parasitism. In comparison to males, female *M. amalphitanum* and *M. caribea* have two more types of sensilla (BS and PS).

We found no correlation between the number of antennal sensilla types and body size in our data or when these were combined with the data on larger Chalcidoidea and Braconidae (Fig. 7B, Table 2). The need for heterogeneous and varied information about the environment necessitates a diversity of receptors to detect that information, no matter what the size of the organism.

The possible functions of the sensilla in our work have been derived from morphological features, such as the overall shape of sensillum, presence and number of pores, presence of a basal socket, and position of the sensillum on the antenna (*Van Baaren et al., 2007*). Comparison with earlier data was based on the external morphology of sensilla as seen in SEM photographs.

### Sensilla chaetica, aporous (ChS-AP)

ChS-AP are common in parasitoid wasps and have been described in almost every study on antennal sensilla. They often are the most abundant type on the antenna (*Onagbola &*

*Fadamiro, 2008*; *Das et al., 2011*; *Xi et al., 2011*; *Zhou et al., 2011*; *Zhou et al., 2013b*; *Yang et al., 2016*). Their aporous walls, elongated shape, and presence of the basal socket suggest a mechanosensory function, which agrees with TEM data (*Chiappini, Solinas & Solinas, 2001*).

### Sensilla trichodea, aporous type 1 (TS1-AP)

These sensilla are the most abundant in the studied species. They were described in species of *Trichogramma* (*Olson & Andow, 1993*; *Amornsak, Cribb & Gordh, 1998*; *Consoli, Kitajima & Parra, 1999*; *Zhang et al., 2012*; *Van der Woude & Smid, 2016*) and one species of Braconidae (*Dweck, Gadallah & Darwish, 2008*). TS1-AP lack pores and a basal socket and therefore were considered in most earlier studies as protective structures without sensory function, and described as microtrichia or setiform structures rather than sensilla. However, their inner ultrastructure has not been described yet, and thus their true function remains unknown.

### Sensilla trichodea, aporous type 2 (TS2-AP)

TS2-AP are typical for parasitoid wasps; they appear in functional groups at the head-scape and scape-pedicel joints and are sometimes called "hair plates" (*Consoli, Parra & Zucchi, 2010*). They have been shown to measure the relative position of the scape with respect to the head in *Trichogramma minutum* Riley, 1871 (Hymenoptera: Trichogrammatidae). When the wasp moves its antenna, these sensilla touch the surface of antennal socket, and their initial orientation is distorted (*Schmidt & Smith, 1986*). Thus, these sensilla are considered proprioceptors (*Onagbola & Fadamiro, 2008*; *Dweck, 2009*; *Xi et al., 2010*; *Xi et al., 2011*; *Zhou et al., 2011*; *Zhou et al., 2013b*; *Zhou et al., 2013a*; *Ahmed et al., 2013*; *Namikawa & Amornsak, 2016*; *Yang et al., 2016*). These sensilla are sometimes termed the "Böhm bristles" after Böhm, who described them in Lepidoptera (*Böhm, 1911*).

### Sensilla trichodea, uniporous (TS-UP)

These sensilla appear unique to the studied species; sensilla similar to TS-UP in shape were described in *Trichogramma nubilae*, *T. galloi* and *T. pretiosum* (*Olson & Andow, 1993*; *Consoli, Kitajima & Parra, 1999*); however, they were multiporous and lacked a small expansion with an apical pore at the tip. The absence of wall pores and a basal socket together with the presence of an apical pore suggests gustatory function for TS-UP. In males, these sensilla are the most apically protruding; they may be used in mating behavior.

### Sensilla styloconica (SS)

SS were not yet reported in parasitoid wasps. However, SS are reminiscent of the frequent coeloconic sensilla, which are described as short pegs set in pits, supposedly thermo/hygroreceptors (*Consoli, Parra & Zucchi, 2010*). Additionally, in *Anagrus atomus* (Linnaeus, 1767) (Hymenoptera: Mymaridae) coeloconic sensilla have exactly the same location as in *M. mymaripenne* and *M. caribea*, i.e., one on the 2nd flagellomere and one on the club (*Chiappini, Solinas & Solinas, 2001*). In insects thermo/gygroreceptors can take different shapes, including styloconic-like sensilla (*Altner & Loftus, 1985*). Thus, we propose thermoreceptive and/or hygroreceptive function for these sensilla.

### Sensilla basiconica (BS)

This robust sensilum, multiporous at its tip, is unique to females in the studied species and appears apically on the club. Basiconic-shaped sensilla with the same features have been reported in *Anaphes victus* Huber, 1997 and *Anaphes listronoti* Huber, 1997 (Hymenoptera: Mymaridae), named sensilla chaetica type 2 by *Van Baaren et al. (1999)*. There are two per female antenna at the distal end of the club, whereas male *A. victus* and *A. listronoti* lack BS entirely. Similar sensilla were also reported for female *A. atomus*, two per antenna, located apically on the club (*Chiappini, Solinas & Solinas, 2001*). In that study, gustatory function was supposed for BS based on the TEM photographs and presence of an apical pore system. The authors proposed involvement of BS in the recognition of microhabitat, which is the host-plant tissue wounded by the ovipositor of the host. This may also be the case in the species studied by us, as their host, the thrips *H. haemorrhoidalis,* inserts its eggs into leaves (*Del Bene, Gargani & Landi, 1998*). Sensilla resembling BS were also described in the Braconidae species *M. croceipes* and *C. marginiventris*. The overall composition and multiporous tip are almost identical to BS, but these sensilla have longitudinal grooves, are not exclusively female, and are relatively abundant (*Das et al., 2011*). In *Metaphycus parasaissetiae* (Hymenoptera: Encyrtidae) basiconic-shaped sensilla also appear only at the apex of the club, but are uniporous at the tip; they are abundant, and are found in both males and females (*Zhou et al., 2013b*).

### Multiporous placoid sensilla (MPS)

MPS are common in parasitoid wasps and have been described in Chalcidoidea and larger Braconidae as elongated sensilla fused with the antennal surface and covered with numerous pores (*Van Baaren et al., 1999*; *Xi et al., 2010*). MPS of *Megaphragma* species demonstrate a unique structure, the elongated projection at the tip (Fig. 3A); MPS with a similar but not identical tip shape were described in *Trichogramma galloi*, *T. pretiosum,* and *T. evanescens* (*Consoli, Kitajima & Parra, 1999*; *Van der Woude & Smid, 2016*). Such sensilla were shown to have olfactory function in *Apis mellifera* Linnaeus, 1758 (Hymenoptera: Apidae) (*Lacher, 1964*). In parasitoid wasps they are possibly responsible for host search and host recognition in females and search for sexual partner in males. Since these functions remain fulfilled in smaller parasitoid wasps, while the number and sizes of MPS are considerably reduced (see corresponding paragraphs of "Discussion"), we suppose that the innervation of MPS should be denser in smaller species. The available data on MPS innervation in parasitoid wasps supports this hypothesis. In larger species of Braconidae and Icheumonidae, such as *Coeloides brunneri* Viereck, 1911 (Hymenoptera: Braconidae) and *Itoplectis conquisitor* (Say, 1835) (Hymenoptera: Icheumonidae), MPS are innervated by 13 and 27 neurons respectively, whereas in the small chalcidoids *Tetrastichus hagenowii* (Ratzeburg, 1852) (Hymenoptera: Eulophidae) and *Torymus warreni* (Cockerell, 1911) (Hymenoptera: Torymidae) more than 50 neurons innervate each MPS (*Richerson, Borden & Hollingdale, 1972*; *Borden, Chong & Rose, 1978*; *Barlin & Vinson, 1981*; *Barlin, Vinson & Piper, 1981*).

### Placoid sensilla (PS)

These large elongated sensilla lack pores or basal socket and appear to be immobile. Thus, PS are neither mechanosensors nor chemoreceptors. No such structures have been previously

described in parasitoid wasps and their function is unknown. Yet they are present in females of all three studied species, which implies their importance for the biological tasks of the female in *Megaphragma*.

## Distribution

Within each of the species studied, sensilla distribution was found to be invariable. Between the studied species, significant differences were observed only on the flagellum so the scape and pedicel are omitted in Fig. 6, showing the distribution of antennal sensilla in *M. mymaripenne* and *M. caribea*, since these segments are identical in all three studied species. Terminal flagellar segments appear to be the most species-specific parts of the antenna in Chalcidoidea (*Barlin, Vinson & Piper, 1981*). Constant sensilla distribution within species and constancy of interspecies differences make the distribution of antennal sensilla a reliable distinguishing trait for species identification.

## Sizes of antennal sensilla

The antennal sensillum sizes vary moderately between the studied species (Table 1, Table S3). It has been found that the types of sensilla which appear on several flagellomeres (such as ChS-AP, TS1-AP, TS2-AP and SS) tend to differ considerably in length and, less often, in diameter between different flagellomeres in specimens of the same species and sex (Tables S2–S4). Therefore, interspecies and intersex comparisons for these types of sensilla have been made separately for each flagellomere.

Females tend to have considerably longer and wider sensilla than males in both *M. amalphitanum* and *M. caribea* (Tables S2 and S3). In three comparable studies, treating *Lysiphlebus fabarum* (Marshall, 1986), *O. phongi,* and *T. australicum,* where measurements were provided separately for each sex (*Amornsak, Cribb & Gordh, 1998*; *Xi et al., 2010*; *Xi et al., 2011*), no such clear tendency in size among specimens of the same species and sex was observed. In other detailed studies on the antennal sensilla of parasitoid wasps the data on the sizes of sensilla were not specified between sexes for most types of sensilla.

In both sexes *M. amalphitanum* have sensilla longer than in *M. caribea*, while *M. mymaripenne* tend to have medium sensillum length. Sensillum diameters are prone to vary insignificantly, especially in males (Tables S1 and S2).

The data on three types of sensilla, chemoreceptors (MPS), mechanoreceptors (ChS-AP) and proprioceptors (TS2-AP) in the studied species were combined with data on larger parasitoid wasps. Possible homologues of these sensillum types in each study have been deduced from morphological traits, relative positions, and distribution of sensilla.

Only TS2-AP length and width have been found to decrease significantly with the body size (Figs. 8A and 8B; Table 2). TS2-AP have been shown to inform parasitoid wasps on the relative position of their antenna through changes in the scapal-head angle: TS2-AP on the scape touch the socket of antenna as it moves and deviate to different degrees from their initial orientations (*Schmidt & Smith, 1986*; *Consoli, Parra & Zucchi, 2010*). It seems that the same mechanism is implemented in the scape-pedicel joint, the second location of these sensilla. Thus, the sizes of TS2-AP should depend on the size of the antennal socket and the distance between the scape and pedicel, which decrease with the body size. Additionally,

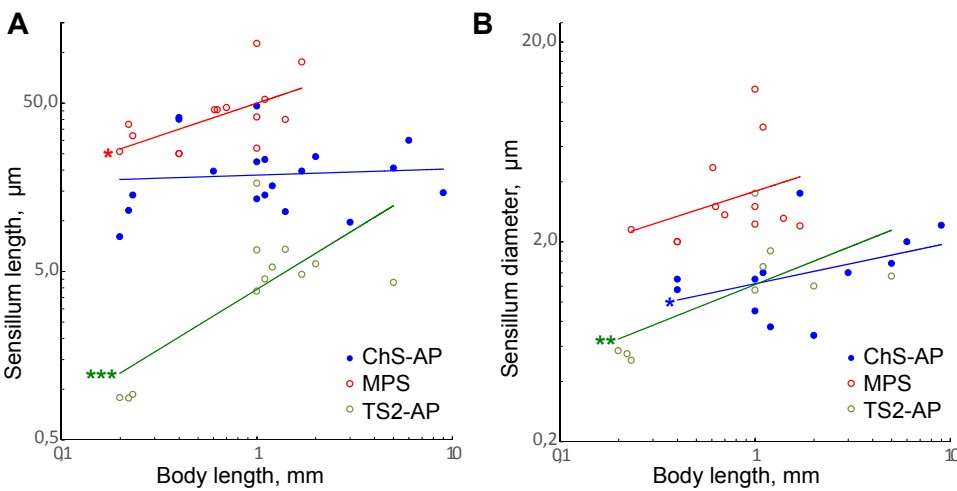

**Figure 8** **Size changes of three antennal sensilla types, mechanoreceptive ChS-AP, chemoreceptive MPS, and proprioreceptive TS2-AP in parasitoid wasps.** (A) Sensillum length as a function of body length. (B) Sensillum diameter as a function of body length. Data on Chalcidoidea, Scelionidae and Braconidae species are included. For references, see Table S4. Trendlines representing correlated variables with the slopes of SMA and OLS regressions significantly different from zero are marked with three asterisks (***) for $p < 0.01$ and two asterisks (**) for $0.01 \leq p < 0.05$. Trendlines for which only OLS demonstrated correlation between variables and significant difference of slope from 0 are marked with one asterisk (*) for $0.01 \leq p < 0.05$ (for slope values and $R^2$ see Table 2).

TS2-AP appear to be remarkably shorter in *Megaphragma* species than in larger parasitoid wasps, in which they are 4–17 times longer.

MPS length and ChS-AP shows correlation between variables and difference of slope from zero only in OLS regression analysis. Also, the regression slope values were relatively small (0.39 and 0.25, respectively). Therefore, we suppose that either there is only weak correlation of these parameters with the body length, or that it is absent.

## Miniaturization of sense organs in parasitoid wasps

Our results demonstrate a strong correlation between the number of antennal sensilla and the body size in parasitoid wasps. These findings are comparable to the findings on insect compound eyes, as the decrease in body size also leads to a considerable decrease in the number of ommatidia (*Polilov, 2016*). Thus, species of genus *Megaphragma* have about 30 ommatidia per eye (*M. carribea* 32 ± 3; *M. mymaripenne* 29 ± 1; *M. amalphitanum* 29 ± 1). Ommatidia of the minute *T. evanescens* have inner ultrastructure comparable to *Megaphragma* (*Makarova, Polilov & Fischer, 2015*), however, the number of ommatidia in their eye is about 4.5 times as high (*Fischer, Müller & Meyer-Rochow, 2011*).

Sense organ sensitivity in insects with moderate differences in body size is affected mostly by their lifestyles (*Stöckl et al., 2016*). Insects can have a 30 times difference in body size and 1,000 times difference in the body volume; it seems logical to suggest that the difference in the sensitivity of their sensory organs should also be considerable though there is no definite evidence for this. While relatively reduced sensitivity might be true for vision (*Palavalli-Nettimi & Narendra, 2018*), *Megaphragma* species seem to have a level of

olfactory and gustatory sensitivity close to that of larger parasitoid wasps. This needs to be further investigated to make more accurate statements about the retention of sensitivity in a miniaturized antenna.

The miniaturization of insect sensory organs results in the reduction of the relative volume of the structures which process sensory information (*Makarova & Polilov, 2013a*), e.g., the complexity of antennal sensory system was shown to correlate with the size of antennal lobes and individual glomeruli (*Kelber, Rössler & Kleineidam, 2009*; *Mysore, Shyamala & Rodrigues, 2010*), while the number of ommatidia in a compound eye was shown to correlate with the size of the optic lobes (*Power, 1943*; *Rein, Zöckler & Heisenberg, 1999*; *Gronenberg & Hölldobler, 1999*). Interestingly, the decrease of the relative volume was shown to be different for each of the optic ganglia (*Makarova & Polilov, 2013b*).

Despite a dramatic reduction in the number of antennal sensilla and number of ommatidia in miniaturized parasitoid wasps, the sizes of these sense structures did not show any strict correlation with the body size. It appears that the limit of eye miniaturization is set by the diameter of one facet, which determines the power of the facet lens. The smallest facet diameter in insects was recorded in *T. evanescens* ($6.39 \pm 0.33$ μm; *Fischer, Müller & Meyer-Rochow, 2011*), while in smaller *Megaphragma* it was measured as $8.1 \pm 0.3$ μm (*Makarova, Polilov & Fischer, 2015*). Antennal sensilla sizes are also mostly uncorrelated with the body size in parasitoid wasps. However, it seems that *Megaphragma* have larger relative sensilla diameters than most other chalcidoids (from 0.32 μm (TS2-AP) to 2.6 μm (TS1-AP)). It appears that sensilla and ommatidia are hard to miniaturize because of the complexity of their ultrastructure, which imposes the limits of their reduction.

## CONCLUSIONS

The antennae of *Megaphragma* are the smallest functioning parasitoid antennae described to date. Large-scale comparative analysis of antennal sensilla of *Megaphragma* with antennal sensilla of larger Chalcidoidea and Ichneumonoidea demonstrated that such an extreme miniaturization resulted in a significant decrease of number of antennal sensilla, with only 39–49 antennal sensilla remaining. However, the decrease in the body size did not affect the number of antennal sensilla types. The studied species have eight types of antennal sensilla, including two (SS and PS) not described previously, while some of the larger species have only four. The individual sensilla sizes were also almost unaffected by the changed body size. A reduction of the number of functional elements in a sense organ accompanied with a minor decrease in their sizes is a common pattern in miniaturized insects. We suppose that the complexity of the inner ultrastructure of the sensilla prevents further miniaturization of antennal sensilla, with the smallest sensillum observed being only 0.48 μm in length and 0.32 μm in diameter. Thus, it is very likely that *Megaphragma* species are close to the limit of possible reduction in a functional antenna required for the parasitoid lifestyle.

Composition and shape of the antennae along with the overall number, distribution and relative position of antennal sensilla were invariable between specimens of the same sex and species. These traits may be potentially used as reliable distinguishing characters in the taxonomy of miniature parasitoid wasps.

## ACKNOWLEDGEMENTS

We are grateful to Petr Petrov for helpful comments on the manuscript.

### Funding

This work was supported by the Russian Science Foundation (project no. 17-74-10246 to Anastasia A. Makarova and project no. 14-14-00208 to Alexey A. Polilov). The funders had no role in study design, data collection and analysis, decision to publish, or preparation of the manuscript.

### Grant Disclosures

The following grant information was disclosed by the authors:
Russian Science Foundation: 17-74-10246, 14-14-00208.

### Competing Interests

The authors declare there are no competing interests.

### Author Contributions

- Anna V. Diakova performed the experiments, analyzed the data, prepared figures and/or tables, authored or reviewed drafts of the paper, approved the final draft.
- Anastasia A. Makarova performed the experiments, approved the final draft.
- Alexey A. Polilov conceived and designed the experiments, performed the experiments, analyzed the data, contributed reagents/materials/analysis tools, approved the final draft.

### Data Availability

   The raw data is included in the Supplemental Files.

### Supplemental Information

Supplemental information for this article can be found online at http://dx.doi.org/10.7717/peerj.6005#supplemental-information.

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
