# Peer review of "Between extreme simplification and ideal optimization: antennal sensilla morphology of miniaturized Megaphragma wasps (Hymenoptera: Trichogrammatidae)"

_PeerJ, doi:10.7717/peerj.6005_

## Round 0.1 · original submission · Minor Revisions

Dear Dr. Diakova and colleagues:

Thanks for submitting your manuscript to PeerJ. I have now received two independent reviews of your work, and as you will see, both are very favorable. Well done! Nonetheless, both reviewers raised some relatively minor concerns about the research, and areas where the manuscript can be improved. A few things in particular: 1) make sure missing references are included in your work, 2) make sure that ALL figure legends contain all of the information necessary to make the figure “stand alone”, 3) try to provide images that more clearly accentuate the structures that you are observing. Please also remember that both reviewers provided marked-up versions of your manuscript.

I agree with the issues raised by the reviewers, and thus feel that their concerns should be adequately addressed before moving forward.

Therefore, I am recommending that you revise your manuscript accordingly, taking into account all of the issues raised by the reviewers. I do believe that your manuscript will be ready for publication once these issues are addressed.

Good luck with your revision,

-joe

Reviewer 1 ·

Basic reporting

pdf attached

Experimental design

pdf attached

Validity of the findings

pdf attached

Annotated reviews are not available for download in order to protect the identity of reviewers who chose to remain anonymous.

Reviewer 2 ·

Basic reporting

no comment

Experimental design

no comment

Validity of the findings

no comment

Additional comments

The paper is written in a good English, although I’m not a native English speaker. The structure of the paper is good and meets the Journal criteria and needs. References are not accurate either in terms of year of citation or journal. I noticed two mistakes (see attached pdf), there could be more but I’m not going to check all the references, this has to be carefully done by the authors. Legend of figures are very scarce and not informative at all, they should describe what’s in the plates and schemes much more in details, especially when the pictures show an extensive lettering (i.e. fig3). I found Fig 3 rather confused hard to read, I suggest to split it into 2 plates of the same size incorporating more data on the reported structures (details of sockets, structures on males). The authors reported data of three Megaphragma species, in two species of both sexes. I suggest the authors to add SEM data of males as well, Fig2 is just not enough to pick the differences and drawings are not true data.
I would like to see more pictures of the multiporous tip of the basiconic sensillum, it is not clear to me on the basis of the picture in Fig.3.
I’m also rather convinced that PS is an olfactory placoid sensillum, please provide pictures clearly showing the absence of pores (130.000 x is not needed).
Comments and suggestions are reported on the attached PDF.

Annotated reviews are not available for download in order to protect the identity of reviewers who chose to remain anonymous.

---

## Round 0.2 · accepted · Accept

Dear Dr. Diakova and colleagues:

Thanks for re-submitting your manuscript to PeerJ, and for addressing the concerns raised by the reviewers. I now believe that your manuscript is suitable for publication. Congratulations! I look forward to seeing this work in print, and I anticipate it being an important resource for the hymenopteran community, especially researchers interested in trichogrammatid systematics. Thanks again for choosing PeerJ to publish such important work.

Best,

-joe

#